Acute toxicity and responses of antioxidant systems to dibutyl phthalate in neonate and adult Daphnia magna

Shen Chenchen
Wei Jie weijiedlou@foxmail.com
Wang Tianyi
http://orcid.org/0000-0001-8407-635X Wang Yuan wangyuan@dlou.edu.cn
Key Laboratory of Hydrobiology in Liaoning Province, Dalian Ocean University , Dalian , China
Anderson Todd
Electronic publication date: 2019 Mar 14
Publication date: 2019
Volume: 7
Electronic Location ID: e6584
Received 2018 Nov 25; Accepted 2019 Feb 6
Copyright: © 2019 Shen et al.
Copyright year: 2019
Copyright holder: Shen et al.
License: This is an open access article distributed under the terms of the Creative Commons Attribution License, which permits unrestricted use, distribution, reproduction and adaptation in any medium and for any purpose provided that it is properly attributed. For attribution, the original author(s), title, publication source (PeerJ) and either DOI or URL of the article must be cited.
License URL: https://creativecommons.org/licenses/by/4.0/

Keywords: Acute toxicity, Antioxidant enzymes, Lipid peroxidation, Daphnia magna, Dibutyl phthalate

Funding: National Natural Science Fund of China 41501535 Natural Science Fund of Liaoning Province 20180550774 This work was supported by the National Natural Science Fund of China (No. 41501535), and the Natural Science Fund of Liaoning Province (No. 20180550774). The funders had no role in study design, data collection and analysis, decision to publish, or preparation of the manuscript.

==============================
Dibutyl phthalate (DBP) poses a severe threat to aquatic ecosystems, introducing hazards to both aquatic species and human health. The ecotoxic effects of DBP on aquatic organisms have not been fully investigated. This study investigates acute toxicity, oxidative damage, and antioxidant enzyme parameters in neonate and adult Daphnia magna exposed to DBP. The obtained results show comparable DBP toxic responses in neonates and adults. The median lethal concentrations (LC50) of DBP in neonates exposed for 24 and 48 h were 3.48 and 2.83 mg/L, respectively. The LC50 of adults for the same DBP exposure durations were 4.92 and 4.31 mg/L, respectively. Increased hydrogen peroxide and malondialdehyde were found in neonates and adults at both 24 and 48 h, while the total antioxidant capacity decreased. Superoxide dismutase activity increased significantly in neonates and adults exposed to 0.5 mg/L DBP, and subsequently diminished at higher DBP concentrations and prolonged exposure. Catalase and glutathione S-transferases activities both decreased markedly in neonates and adults. The changes observed were found to be time and concentration dependent. Overall, these data indicated that the acute toxic effects of DBP exposure on neonates were more pronounced than in adults, and oxidative injury may be the main mechanism of DBP toxicity. These results provide a functional link for lipid peroxidation, antioxidant capacity, and antioxidant enzyme levels in the Daphnia magna response to DBP exposure.

Introduction

Phthalate esters (PAEs) have been extensively used as plasticizers for polymers, despite their established action as endocrine-disrupting chemicals and their severe impact on human health (Daiem et al., 2012; Fernández, Gómara & González, 2012). The annual production capacity of PAEs exceeds approximately 8 million tons globally (Wittassek et al., 2011). PAEs can be found in various products, such as toys, personal-care products, pharmaceuticals, medical tubes and devices, packaging, detergents, and food items. However, since PAEs are non-covalently bound to polymer matrices, they can easily disperse and contaminate aquatic ecosystems, via discharges of industrial and agricultural wastewater, sewage sludge, and incorrect disposal of plastic items (Daiem et al., 2012; Zhang et al., 2018).

Dibutyl phthalate (DBP) is the most frequently reported PAE pollutant, and has been detected in a variety of water sources and sediments (Huang, Sun & Song, 1999). High levels of DBP were reported in the South African Songhua River (3.42–10.2 mg/L) (Fatoki et al., 2010). In China, DBP concentrations detected in the lakes of urban Guangzhou range from 0.94 to 3.60 mg/L (Zeng et al., 2009). DBP was also the primary pollutant that was detected in a water environment survey of the Yangtze River. Especially, in the Wuhan section, the concentrations of DBP in most of the tested samples exceeded the national water quality criteria (3.0 μg/L) (Fan, Xia & Sha, 2008; Wang, Xia & Sha, 2008b). DBP is susceptible to hydrolysis in aquatic ecosystems at a slower rate (Staples et al., 1997b; Thomsen, Rasmussen & Carlsen, 1999). The photooxidation half-life for DBP in water has been estimated at 144 days (Staples et al., 1997a). It has been reported that the biodegradation half-life of DBP in natural surface water is 1–14 days, and the biodegradation half-life in groundwater is 2–23 days (Staples et al., 1997b; Thomsen, Rasmussen & Carlsen, 1999). Although the utilization of DBP as a plasticizer has been banned in the United States, the United Kingdom, and China, it can still be found in aquatic ecosystems (Lin et al., 2018). The minimum toxic effect concentration of DBP for aquatic species is 100 μg/L (Staples et al., 1997a). However, monitoring data in specific areas either reached or exceeded this concentration (Fatoki et al., 2010; Zeng et al., 2009). Aquatic species are continuously exposed to DBP, which affects the health of aquatic organisms through various mechanisms and damages aquatic ecosystems (Brown et al., 1998; Staples et al., 1997a; Gu et al., 2017).

Previous risk assessment reports have focused on the detrimental effects of DBP on aquatic animals (Brown et al., 1998; Agus et al., 2015; Li et al., 2015) DBP toxicity is partly a result of its lipophilic nature, resulting in ease of accumulation in animal tissues (Agus et al., 2015; He et al., 2011). DBP has been found to induce acute and chronic toxicity in various aquatic species, such as the yellow perch Perca flavescens, the rainbow trout Oncorhynchus mykiss, the nile tilapia Oreochromis niloticus, the mirror carp Cyprinus carpio, and the red killi fish (Staples et al., 1997a; Agus et al., 2015; Khalil, Elhakim & EL-Murr, 2016). A key detrimental physiological effect of DBP exposure in aquatic organisms is the induction of growth disorders (Janjua et al., 2007), as well as the disruption of both development (Xu et al., 2015) and reproduction (Lu, Lin & Aitken, 2017). DBP accumulates within organism tissues where it can alter and degrade enzymatic processes, which may lead to cell death (Zhao, Gao & Qi, 2014). Moreover, DBP has been found to promote oxidative damage in the dinoflagellate Gymnodinium breve due to elevated intracellular levels of reactive oxygen species (ROS) (Bie et al., 2012). Accumulated active oxygen oxidize biomacromolecules include nucleic acids, proteins, and lipid, and result in metabolic abnormalities in biochemical reactions in vivo (Hamed et al., 2017). Excessive ROS can subject biological membranes to oxidation, thus attacking their cellular lipid peroxidation. High DBP concentrations were reported to cause oxidative injury and elevated malondialdehyde (MDA) levels in the red tide dinoflagellate Karenia brevis (Li et al., 2015). To maintain the body’s homeostasis and prevent further oxidative injury induced by toxic substances, many organisms have evolved antioxidant defenses (Sen et al., 2010). Active oxygen scavenging systems play a central role in these protective defense responses. ROS are abolished by superoxide dismutase (SOD), catalase (CAT), glutathione peroxidase (GPx), glutathione S-transferase (GST), and non-enzymatic scavengers (Xing et al., 2012). These systems protect cells from oxidative injury caused by a range of stresses, such as exposure to DBP toxicity (Li et al., 2015; Zhao, Gao & Qi, 2014). A previous publication reported that under DBP stress conditions, accumulation of SOD results in the scavenging of ROS in Oreochromis niloticus (Khalil, Elhakim & EL-Murr, 2016). DBP stimulated CAT and GSH activities in the flounder Paralichthys olivaceus, thus counteracting the oxidative stress (Kang et al., 2010). A study on the goldfish Carassius auratus (Zhao, Gao & Qi, 2014) showed that oxidative damage was detected in all investigated tissues after 96 h of DBP exposure. Oxidation-related biomarkers are currently extensively employed to assess exposure levels and harmful effects of pollutants on aquatic organisms (Lacroix et al., 2015; Huang, Li & Yang, 2016).

Daphnia magna is an extensively located zooplankton species and a critical link in the food webs of freshwater ecosystems. They are a key primary consumer and are a primary food source for various planktivorous fish species (Galdiero et al., 2017). Accordingly, Daphnia are not only a commonly used model organism for toxicological surveys of aquatic ecosystem, but are also considered a valuable indicator species for environmental pollution assessments (Cui, Chae & An, 2017). A previous evaluation of the toxicity effects of phthalates showed that diethylhexyl phthalate (DEHP), DBP, and diethyl phthalate (DEP) had detrimental impacts on the fat metabolism, development, reproduction and lifespan of Daphnia magna (Seyoum & Pradhan, 2019). Huang et al. (1998) reported a median lethal concentration (LC50) for Daphnia magna under DBP treatment at 24 h of 8.0 mg/L. The LC50 of DBP, DEHP, and DEP for Daphnia magna exposed for 48 h were reported to be 6.78, 0.003, and 90.0 mg/L, respectively (Jonsson & Baun, 2003). Furthermore, a previous study reported that exposure to 1.8 mg/L of DBP caused a significant decline in the reproductive rate of Daphnia magna (Mccarthy & Whitmore, 1985). A recent study also investigated independent and co-toxicity mechanisms in both bacteria Photobacterium phosphoreum and Daphnia magna exposed to DBP and copper (Huang, Li & Yang, 2016). We recently reported that the cat and gst gene expression levels were notably reduced or increased upon DEHP exposure in Daphnia magna (Wang et al., 2018). Organisms in the early developmental stages have been reported to be less tolerant to the detrimental effects of pollutants (Huang, Li & Yang, 2016). Many recent publications showed the teratogenic nature of DBP, reporting malformations in developing Zebrafish Danio rerio, the African clawed frog Xenopus laevis, and embryos of the polychaete Galeolaria caespitosa (Xu et al., 2015; Gardner et al., 2016; Lu, Lin & Aitken, 2017). Environmental exposure may lead to DBP accumulation during the neonatal stage, which negatively impacts various future developmental characteristics (Khalil, Elhakim & EL-Murr, 2016). Until now, little is known regarding the differences in the deleterious effects of DBP on Daphnia magna in neonatal and adult antioxidant defense systems. Therefore, in parallel tests, this study assessed the sensitivity of Daphnia magna in neonates and adults exposed to the toxic pollutant DBP.

The objectives of the present work are (1) to compare the acute toxicity of Daphnia magna in neonates and adults exposed to DBP, (2) to establish safe concentration (SC) of DBP exposure to neonatal and adult Daphnia magna, for the application of risk evaluations of polluted fresh water, (3) to further investigate the mechanism of DBP toxicity by determining its effects on lipid peroxidation and oxidative damage in both neonatal and adult Daphnia magna populations. These data provide a reference for the further ecological risk evaluation of PAE exposure of aquatic organisms.

Materials and Methods

Experimental materials

Daphnia magna were provided by the Key Laboratory of Hydrobiology at Dalian Ocean University (Liaoning Province, China). Neonates or adults that were synchronously released from laboratory-cultured clonal line of Daphnia magna were used for the experiments. The clonal line was initiated with an individual, parthenogenetic female. The neonates produced by this first female were selected and then cultivated. Experiments were performed using four to six brooding offspring. The selected neonates or adults were cultured in dechlorinated water in a 500 mL glass beaker, kept in an illumination incubator (Thermo Fisher, Dreieich, Germany). The incubation conditions were adjusted to 22 ± 2 °C on an alternate 12 h light-dark cycle. The light intensity parameter of the experimental animal culture was set to 2,000 lux. The total hardness, alkalinity, dissolved oxygen, and pH of culture water, were 7.29 ± 0.1 mmol/L, 12.48 ± 0.4 mmol/L, 7.96 ± 0.3 mg/L, and 7.8 ± 0.2, respectively. Concentrated Scenedesmus sp. cultures were utilized as food for Daphnia magna at a feeding density of 3 × 105–4 × 105 cell/mL.

Dibutyl phthalate (CAS No: 84-74-2) was bought from Sinopharm Chemical Reagent Co., Ltd. (Shanghai, China), and was reconstituted with acetone for toxicity experiments.

Acute toxicity assay

The toxicity of DBP was determined after 24 or 48 h of static exposure via acute toxicity test published by OECD (2004). Newly hatched neonates (length 0.86 ± 0.08 mm) and adults (length 2.67 ± 0.05 mm) were used for the acute toxicity test. The test concentrations of DBP were established based on the data obtained by pre-experimentation and previously reported literature (Adams et al., 1995). DBP concentrations were set to one, two, three, four, and five mg/L. Culture media without added DBP and solvent was used as blank control. A solvent control (0.008% acetone only treatment, v/v) was also included to determine baseline toxicity. Preliminary experiments and previous literatures confirmed that acetone at 0.008% had no effect on survival or biochemical responses in Daphnia magna (Sancho et al., 2009; Zhao, Gao & Qi, 2014). Five replicates were conducted per treatment concentration (blank control, solvent control, plus DBP concentrations). For each replicate, 10 Daphnia magna neonates or adults were cultured in a glass beaker containing 100 mL of DBP solutions or control solutions and 50 organisms were used for each different concentration of DBP. Neither Daphnia magna neonates nor adults were fed during the test. The movement status of Daphnia magna was examined 24 or 48 h post DBP exposure. Daphnia magna were regarded as immobile if they did not swim 15 s post gentle agitation.

The test compound DBP concentrations were determined by GC/MS mass spectrometry. Nominal and measured concentrations of DBP are presented in Table S1. The stability of DBP in different media has been reported previously (Bajt, Mailhot & Bolte, 2001; Zhou et al., 2005; Wang, Wang & Fan, 2008a). LC50 was defined as the concentration of a substance that induces mortality in 50% of the test group in a short-term experiment. The SC of the substance was calculated via Eq. (1) (Sprague, 1971):(1) SC=24hLC50×0.3(24hLC50/48hLC50)3

Sublethal toxicity bioassays

After LC50 values were calculated, Daphnia magna were exposed to sublethal toxic concentrations of DBP to test both MDA content and antioxidant enzyme activity. Two DBP concentrations (0.5 and 2.0 mg/L) were chosen, and both blank control (no DBP no acetone) and solvent control (0.008% acetone, v/v) were included to determine the baseline reads on enzyme activity assays. For each time endpoint (0, 24, and 48 h), a corresponding control was performed at the same growth stage as the treatment groups. Each group contained four replicates. For each replicate, 500 neonates were randomly grouped to a beaker containing five L of test solution, and 300 adults were also placed in a different five L glass beaker. Both the neonates and adults were exposed under standard culture conditions without feeding throughout the experiment duration. At 24 or 48 h post-exposure, 200 living organisms were harvested for analysis as one replicate. The collected live Daphnia magna were first rinsed with distilled water three times prior to pooling in centrifuge tubes (1.5 mL). All samples used for MDA and enzyme activity analysis were either kept on ice throughout the experiment or at −20 °C for long term storage.

Following DBP exposure, each sample was homogenized at a 10% (w/v) dilution in 1× PBS, and then, the supernatants were obtained after centrifugation at 10,000 r/min for 10 min by Centrifuge 5,804 (Eppendorf, Billerica, Massachusetts, USA). The protein content, total antioxidant capacity (T-AOC), MDA, hydrogen peroxide (H2O2), and enzyme activity levels were measured with commercially available kits (Nanjing Jiancheng Institute, Nanjing, China).

Oxidative stress assays (biomarkers)

To analyze the oxidative stress status of Daphnia magna, the contents of MDA and H2O2 were measured. The protein content was detected with the Bradford method using BSA as protein standard (Bradford, 1976). H2O2 can be reacted with a chromogenic agent to produce a molybdenic acid-peroxide complex. H2O2 was detected by measuring the absorbance of the molybdenic acid-peroxide complex at 405 nm using a H2O2 assay kit (Jiang, Woollard & Wolff, 1990). MDA as the final lipid peroxidation product was determined based on the formation of a MDA/thiobarbituric acid (TBA) complex (Dhindsa, Plumbdhindsa & Thorpe, 1981). A total of 1.5 mL of Daphnia magna was homogenized with an Ultra-Turrax (IKA) in one ml 0.02% butylhydroxytoluene methanol solution in Milli-Q water. Then, three ml of 1% phosphoric acid and one ml of 1% TBA solution were added. Mixtures were mixed, incubated for 30 min at 100 °C, and then cooled on ice for 10 min before addition of five ml butanol. Separation of the organic phase was performed by centrifugation at 10,000 rpm for 15 min at 4 °C. The supernatant was taken and measured at 532 nm. The obtained results were presented as nmol/mg protein. Absorbance values were assessed by spectrophotometer (Thermo Fisher Instruments Inc., Vantaa, Finland).

Measurements of antioxidant parameters

Total antioxidant capacity was measured as the deoxidation ability of Fe3+ to Fe2+ following the principles reported by Opara et al. (1999) and were quantified as U/mg protein (each U equals the nmol Fe3+ deoxidated per min per mg protein).

Superoxide dismutase (EC 1.15.1.1) activity was calculated via the ability to inhibit cytochrome c reduction (Manduzio et al., 2003). Reduced cytochrome c by O2− caused an increased absorbance at 550 nm, which was used to measure SOD activity. Each one mL of the reaction mixture contains 50 μM hypoxanthine, 5.6 mU xanthine oxidase, 10 μM cytochrome c, 50 mM phosphate buffer, and 0.1 mM EDTA. The enzyme activity was quantified as U/mg protein. Each SOD unit equals the amount of sample that can inhibit the rate of reduction of cytochrome c by 50%.

Catalase (EC 1.11.1.6) activity was assessed following the Aebi method (Aebi, 1984). Briefly, eight μL of extracted protein sample was mixed with 792 mL of reaction buffer (PBS 10 mM, H2O2 10 mM) in a measuring cuvette and the absorbance was measured at 240 nm for 1.5 min. The values from CAT enzymatic assay were quantified as units of CAT activity per mg of protein (U/mg protein). Each unit of CAT was determined as the amount of enzyme decomposing one mmol of H2O2 per second.

Glutathione S-transferase (EC 2.5.1.18) activity was detected by the conjugation of GSH (Habig, Pabst & Jakoby, 1974) using 1-chloro-2, 4-dinitrobenzene (CDNB) (Sigma, St. Louis, MO, USA) as substrate. Briefly, 15 μL of the extracted protein sample was mixed with 200 μL of reaction buffer, containing 200 mM PBS buffer at pH = 7.5, l-glutathione reduced one mM, and CDNB one mM. The absorbance value was measured at 340 nm. One unit of GST was determined as the amount of enzyme capable of catalyzing one μM of CDNB per minute. The GST activity was quantified as μmol/min/mg/protein.

Statistical analysis

Statistical analyses were conducted using SPSS 21. Statistical data were presented as mean values ±standard deviation in all studies if not otherwise specified. The LC50 and the 95% confidence limits were calculated using Probit. The significance of the various parameters was tested using two-way analysis of variance (ANOVA). A p-value of 0.05 was used as cutoff for statistical significance, while a p-value of 0.01 for used to indicate high significance. Duncan’s multiple range test was used to compare significant differences among treatments.

Results

Acute toxic effects of DBP on Daphnia magna

No neonatal and adult mortality were present in controls (deionized water and acetone) after 48-h of testing. Increasing toxicity was observed in neonatal and adult Daphnia magna, with incremental increases in DBP concentration. As shown in Table 1, a notable correlation was found between the logarithm of DBP concentration and Daphnia magna mortality. Under these experimental conditions, the 24 and 48 h-LC50 values for DBP exposure of adult Daphnia magna (4.92 and 4.31 mg/L) were higher than those for neonatal Daphnia magna (3.48 and 2.83 mg/L). This data suggests that DBP exposure was more toxic to neonates than to adults.

Table 1 Acute toxicity responses of neonate and adult D. magna exposed to DBP.

Age group	Time (h)	Regression equation	Correlation coefficient R2	LC50 (mg/L)	95% Confidence interval	Safe concentration (mg/L)	Replicates/trials	
Neonate	24	y = −1.819 + 3.361x	0.998	3.48	3.09–3.99	0.56	5/2	
48	y = −1.414 + 2.523x	0.992	2.83	2.42–3.33	5/2	
Adult	24	y = −2.185 + 3.008x	0.991	4.92	4.22–6.32	0.98	5/2	
48	y = −1.254 + 1.798x	0.972	4.31	3.50–6.03	5/2	
Note:

Accumulated mortality (%) of neonatal and adult D. magna exposed to various DBP concentrations for 24 and 48 h and the LC50 with 95% confidence limits, were calculated by Probit analysis using SPSS 21. In the regression equation, the logarithm of concentration was indicated by X and the mortality of D. magna was indicated by Y.

The influence of DBP on H2O2 content in Daphnia magna

The obtained data showed that DBP was associated with obvious upregulation of H2O2 content in both neonates and adults. H2O2 content in neonates exposed to 0.5 mg/L DBP was significantly increased at both 24 and 48 h (17.7 ± 1.7, 17.5 ± 1.1, p < 0.01). Exposure to 2.0 mg/L DBP resulted in elevated H2O2 content at 24 h (8.5 ± 1.1), followed by a notable increase at 48 h compared to controls (19.8 ± 0.8, p < 0.01, Fig. 1A). Following 0.5 or 2.0 mg/L DBP exposure treatment of Daphnia magna, in adults, the (24 and 48 h) H2O2 contents increased significantly compared to controls (p < 0.05, Fig. 1B). The induction and accumulation of H2O2 was related to the concentration and treatment time of external DBP.

Figure 1 DBP induced change in D. magna H2O2 concentrations.

(A) neonates and (B) adults, following exposure to 0.5 and 2.0 mg/L DBP for 24 and 48 h. Capital letters (A/B/C) indicate the difference between the 0, 0.5, and 2.0 mg/L DBP treatment groups at 48 h. Lowercase letters (a/b/c) indicate the difference between the 0, 0.5, and 2.0 mg/L DBP treatment groups at 24 h. Different letters indicate significant differences between the DBP treatments. The error bars are standard deviation (n = 4).

The influence of DBP on lipid peroxidation in Daphnia magna

Malondialdehyde levels in neonatal Daphnia magna exposed to 0.5 mg/L DBP were notably elevated at 24 and 48 h (15.6 ± 4.7 and 10.9 ± 4.8), compared to controls (7.1 ± 1.2, 2.6 ± 0.6, p < 0.01, Fig. 2A). The treatment of neonates with 2.0 mg/L DBP after 24 and 48 h caused 7.8-fold and 25.6-fold increases in MDA levels, respectively (p < 0.01, Fig. 2A). The impact of DBP exposure on MDA content in adult Daphnia magna is shown in Fig. 2B. Adults exposed to 0.5 mg/L DBP for 24 h showed no change in MDA levels, while exposure for 48 h notably increased levels compared to control (4.9 ± 0.8 vs 2.2 ± 0.4, p < 0.05). Treatment with 2.0 mg/L DBP significantly elevated adult Daphnia magna MDA levels at 24 and 48 h (8.5 ± 1.1 and 8.7 ± 0.3, p < 0.05).

Figure 2 DBP induced lipid peroxidation in D. magna.

(A) neonates and (B) adults, following exposure to 0.5 and 2.0 mg/L DBP for 24 and 48 h. Capital letters (A/B/C) indicate the difference between the 0, 0.5, and 2.0 mg/L DBP treatment groups at 48 h. Lowercase letters (a/b/c) indicate the difference between the 0, 0.5, and 2.0 mg/L DBP treatment groups at 24 h. Different letters indicate significant differences between the DBP treatments. The error bars are standard deviation (n = 4).

The influence of DBP on T-AOC in Daphnia magna

Following exposure to 0.5 mg/L DBP, T-AOC levels were significantly reduced in neonates both at 24 and 48 h compared to control (p < 0.05 and p < 0.01, respectively), while T-AOC were notably decreased after exposure to 2.0 mg/L DBP for 24 and 48 h compared to control (p < 0.01, Fig. 3A). The effect of DBP on T-AOC in adult Daphnia magna is shown in Fig. 3B. Following 0.5 and 2.0 mg/L DBP treatments, T-AOC levels significantly decreased both at 24 and 48 h (p < 0.01).

Figure 3 DBP induced changes of T-AOC in D. magna.

(A) neonates and (B) adults, following exposure to 0.5 and 2.0 mg/L DBP for 24 and 48 h. Capital letters (A/B) indicate the difference between the 0, 0.5, and 2.0 mg/L DBP treatment groups at 48 h. Lowercase letters (a/b) indicate the difference between the 0, 0.5, and 2.0 mg/L DBP treatment groups at 24 h. Different letters indicate significant differences between the DBP treatments. The error bars are standard deviation (n = 4).

DBP induced activity changes of antioxidant enzyme in Daphnia magna

The influence of DBP on SOD activity in neonate Daphnia magna is shown in Fig. 4A, where neonates exposed to 0.5 mg/L DBP exhibited a notably increased SOD activity at 24 h (11.9 ± 0.2), which then significantly decreased at 48 h (4.3 ± 0.2, p < 0.01). Exposure of neonates to 2.0 mg/L DBP for 24 and 48 h resulted in a significant decrease in SOD activity (p < 0.01). The influence of DBP on SOD activity in adult Daphnia magna is shown in Fig. 4B, where exposure to 0.5 mg/L DBP led to a notable elevation in adult Daphnia magna (24 h) SOD activity (p < 0.01). However, by 48 h SOD activity was notably lower compared to controls (p < 0.01). Exposure of adult Daphnia magna (24 and 48 h) to 2.0 mg/L DBP led to a notable reduction in SOD activity compared to control (p < 0.05).

Figure 4 DBP induced changes in SOD activity in D. magna.

(A) neonates and (B) adults, following exposure to 0.5 and 2.0 mg/L DBP for 24 and 48 h. Capital letters (A/B/C) indicate the difference between the 0, 0.5, and 2.0 mg/L DBP treatment groups at 48 h. Lowercase letters (a/b/c) indicate the difference between the 0, 0.5, and 2.0 mg/L DBP treatment groups at 24 h. Different letters indicate significant differences between the DBP treatments. The error bars are standard deviation (n = 4).

Following exposure to 0.5 mg/L DBP, neonates showed significantly decreased CAT activity at 24 and 48 h compared to control (p < 0.01, Fig. 5A). Following 2.0 mg/L DBP exposure treatment on Daphnia magna, neonates (24 h) CAT activity was notably reduced (p < 0.01). However, CAT activity increased in the 48-h exposure group (p < 0.01). The impact of DBP stress on CAT activity for adult Daphnia magna is shown in Fig. 5B. In all adult groups exposed to 0.5 or 2.0 mg/L DBP (24 and 48 h), the levels of CAT decreased in comparison to the controls (p < 0.01, respectively).

Figure 5 DBP induced changes in CAT activity in D. magna.

(A) neonates and (B) adults, following exposure to 0.5 and 2.0 mg/L DBP for 24 and 48 h. Capital letters (A/B/C) indicate the difference between the 0, 0.5, and 2.0 mg/L DBP treatment groups at 48 h. Lowercase letters (a/b/c) indicate the difference between the 0, 0.5, and 2.0 mg/L DBP treatment groups at 24 h. Different letters indicate significant differences between the DBP treatments. The error bars are standard deviation (n = 4).

Following 0.5 mg/L DBP treatment, neonates showed significant increase in levels of GST activity at 24 h, which then significantly decreased in response to prolonged exposure (48 h) in comparison to control (p < 0.01, Fig. 6A). Following exposure to 2.0 mg/L DBP, CAT activity in neonates was clearly reduced at 24 h compared to controls (p < 0.01). No change was found in other neonatal Daphnia magna exposure groups. The impact of DBP stress on GST activity in adult Daphnia magna is shown in Fig. 6B. Following treatment with 0.5 and 2.0 mg/L DBP, adult Daphnia magna (24 and 48 h) GST activities were all significantly reduced compared to control (p < 0.01).

Figure 6 DBP induced change in GST in D. magna.

(A) neonates and (B) adults, following exposure to 0.5 and 2.0 mg/L DBP for 24 and 48 h. Capital letters (A/B/C) indicate the difference between the 0, 0.5, and 2.0 mg/L DBP treatment groups at 48 h. Lowercase letters (a/b/c) indicate the difference between the 0, 0.5, and 2.0 mg/L DBP treatment groups at 24 h. Different letters indicate significant differences between the DBP treatments. The error bars are standard deviation (n = 4).

Discussion

Dibutyl phthalate is a unique anthropogenic contaminant that has been listed as a priority pollutant to the environment according to various international organizations; it is particularly toxic to aquatic ecosystems (Agus et al., 2015; Xu et al., 2015). Aquatic species in the early stages of development are more susceptible to the detrimental impacts of pollutants. In invertebrates, DBP has been found to exert toxic effects, inhibiting animal survival and fertility in Daphnia Moina macrocopa (Wang et al., 2011b), altering embryogenesis and larval development in Danio rerio (Xu et al., 2015), affecting survival rates and enzyme activities in the goldfish Carassius auratus (Qu et al., 2015), and damaging the defense mechanisms in the giant freshwater prawn Macrobrachium rosenbergii (Sung, Kao & Su, 2003). However, studies that assess the impact of DBP stress on neonatal and adult Daphnia magna survival and antioxidant systems have not been reported. The objectives of the present work were to analyze the acute toxicity and the oxidative stress responses of Daphnia magna neonates and adults exposed to DBP.

Acute toxicity of DBP exposure to neonate and adult of Daphnia magna

Dibutyl phthalate is a commonly detected persistent organic pollutant with detrimental effects on numerous organisms in aquatic environments (Daiem et al., 2012; Gu et al., 2017). Previous reports reported that the harmful impacts of PAEs on juveniles showed age-dependent sensitivities. The statistical changes were often significant and adults were less sensitive than youth (Hutchinson, Solbe & Kloepper-Sams, 1998; Tonk et al., 2012). The LC50 of DBP in neonates were 3.48 and 2.83 mg/L in response to exposure for 24 and 48 h, and the LC50 of DBP in adults were 4.92 and 4.31 mg/L after the same stress duration, respectively (Table 1). The data suggests that DBP was more toxic for neonate Daphnia magna than for adults, which was consistent with the previously reported results. The 96 h LC50 value for DBP exposed to Oreochromis niloticus, was 50 mg/L in adults (Benli, Erkmen & Erkoç, 2016) and 11.8 mg/L in neonates (Khalil, Elhakim & EL-Murr, 2016). Exposures to DBP in aquatic organisms ranged from 24 to 96 h and comprised a relatively wide range of LC50 values (102–105 μg/L). The negative impacts of DBP on aquatic organisms appear to be species-dependent. The findings of the current study differ to data reported for Oreochromis niloticus exposed to DBP, where the LC50 was 50 mg/L for 96 h of exposure (Benli, Erkmen & Erkoç, 2016). Exposure to a mixture of DBP and Cu(II) increased the toxicity in Daphnia magna by practically two orders of magnitude in Photobacterium phosphoreum (Huang, Li & Yang, 2016). This notable difference may be the result of variations in species, the development stages of the experimental animals, or other abiotic factors such as dissolved oxygen, salinity, light intensity, and temperature.

Oxidative stress in DBP exposed Daphnia magna

Malondialdehyde, ROS, and T-AOC, are widely used as biomarkers to study the harmful effects of environmental pollutants in aquatic organisms (Li et al., 2015; Xing et al., 2012). The increase in H2O2 was induced via exposure to 0.5 mg/L DBP in neonates, and significant H2O2 overproduction was induced by 2.0 mg/L DBP in neonates and 0.5 mg/L DBP in adults for 24 h (Fig. 1). Previous research confirmed ROS production as one of the consequences of toxic response to PAEs (Zheng, Feng & Dai, 2013). The H2O2 content was increased after DBP exposure (Fig. 1), indicating ROS production and severe oxidative stress in neonates and adults. Monitoring T-AOC responses provides a more general indication of the antioxidative-oxidative status of an organism, than measurement of separate oxidants and/or antioxidants. The obtained results showed that T-AOC levels were notably decreased following DBP stress both in neonates and adults (Fig. 3). Increased ROS may oxidize cellular biomacromolecules (nucleotide, protein, and lipids), resulting in oxidative injury in organisms. Previous publications have confirmed DBP induced lipid peroxidation in Cyprinus carpio, Danio rerio embryos, and Cyprinus carpio, where DBP stimulated lipoxygenase activity and consequently increased lipid peroxidation (Agus et al., 2015; Dong et al., 2018; Zhao, Gao & Qi, 2014). This study analyzed the degree of lipid peroxidation in Daphnia magna by assessing the levels of MDA. Increased MDA levels in neonate Daphnia magna following exposure to 0.5 and 2.0 mg/L DBP for 24 and 48 h, and adult Daphnia magna following exposure to 0.5 mg/L DBP (48 h) and 2.0 mg/L DBP (24 and 48 h), suggests the induction of lipid peroxidation (Fig. 2). The observed increase in MDA levels indicates that the inducible antioxidant capacity in vivo were insufficient to eliminate excessive oxygen radicals, which is supported by the findings of a previously reported study (Agus et al., 2015). The data obtained in this study showed that after 24 and 48 h of DBP exposure, the accumulation of H2O2 and MDA in neonates was significantly higher than in adults. The elevation of MDA was not found in adults following 0.5 mg/L DBP exposure for 24 h (Fig. 2B), suggesting that mature antioxidant systems provide effective protection against low DBP stress. Therefore, this implies that DBP induced different effects due to oxidative injury in neonate and adult Daphnia magna, where neonate Daphnia magna exhibited higher levels of oxidative damage.

Antioxidant parameters in response to DBP stress in Daphnia magna

To eliminate the increased H2O2 and MDA levels and mitigate oxidative injury due to free radicals, organisms initiate emergency mechanisms such as increasing their cellular levels of antioxidant parameters, such as SOD, CAT, and GST, and reducing endogenous antioxidants (Sen et al., 2010). SOD can catalyze the disproportionation reaction to form O2− into H2O2, which is a precursor of the OH radical. SOD activity is commonly classified as the main defense mechanism against oxidative damage in cells caused by increased ROS and lipid peroxidation (Khan, Khan & Sahreen, 2012). SOD activity was elevated in brains of carp and the neotropical fish Hoplias malabaricus induced by heavy metals, synthetic organic pollutants, and biotoxins (Da Silva et al., 2011; Xing et al., 2012). In the present work, DBP exposure (0.5 mg/L) notably increased SOD activity compared to control. Increased SOD activity could be an adaptive response of organisms to environmental stress with the aim to reduce cell damage in Daphnia magna. By using active oxygen scavenging enzymes as defense system, organisms can resist increased levels of O2−. Therefore, increased SOD activity following DBP exposure, may be attributed to the cell’s adaptive defense mechanism to eliminate surplus O2−. CAT is a key enzymatic antioxidant system, which can eliminate H2O2 produced from ROS, catalyzed by SOD, and thus alleviate cell damage (Zhao, Gao & Qi, 2014). Many environmental pollutants, such as DBP, heavy metals, and organic pesticides, induced elevated levels of CAT activity in Cyprinus carpio, carp, and blue mussels (Xing et al., 2012; Zhao, Gao & Qi, 2014; Lacroix et al., 2015). This study showed that the CAT activity in neonates and adults of Daphnia magna significantly reduced in DBP exposure groups. Increased CAT activity is beneficial for the elimination of excess H2O2 and to reduce the oxidative damage to cells (Gehringer et al., 2004). However, when the CAT H2O2 elimination capacity was saturated, excess H2O2 accumulated and CAT activity was restrained, which resulted in cell damage. This study showed that Daphnia magna under DBP stress reduced CAT activity, and significantly increased SOD activity. The activity of CAT was limited by the intracellular excess of O2−, and the presence of H2O2 inhibited the action of SOD (Xiang et al., 2017). Thus, DBP exposure may produce O2−, resulting in the inhibition of CAT activity, which was also found here.

Glutathione S-transferase was found to play a primary regulatory role in the redox biochemical reaction cycle of cells; in particular, its antioxidant activity is mainly deactivated by ROS or peroxidation products. The intracellular free radical H2O2 can be converted into water and molecular oxygen via the synergistic action of the two antioxidant enzymes CAT and GST (Wang et al., 2011a). In this study, GST activity was suppressed in DBP-stressed Daphnia magna. GST activity in Oreochromis niloticus under heavy metal Cu stress was also notably reduced (Atli & Canli, 2010). A possible explanation is that excessive ROS or severe oxidative injury may lead to GSH depletion and/or disrupted synthesis and inhibition of GST activity. The data suggest that oxidative stress was created in DBP exposed Daphnia magna. The MDA content and the levels of antioxidant parameters depended on both the level of DBP-induced stress and contact time. Under a severe stress environment, ROS exceeded the cellular antioxidant defense capacity, resulting in decreased activity levels of antioxidant enzyme. DBP exposure treatment led to the failure of the antioxidant defense system with changes in lipid peroxidation, which is consistent to previous findings (Oliveira et al., 2012).

Conclusions

Acute toxicity assays and biochemical analyses were performed to establish the role of oxidative injury in response to DBP stress in neonate and adult Daphnia magna. The data from acute toxic assays indicates that DBP exposure was more toxic to neonates than to adults, and Daphnia magna in the neonatal stage were less tolerant to the impacts of DBP. Increased MDA production levels and decreased T-AOC levels were observed, suggesting that ROS-induced oxidative damage was a main toxic effect of DBP. All observed indicators in response to contamination stress were time and concentration dependent. The observation of deleterious impacts of DBP exposure at low concentrations, poses a significant potential risk to the global environmental health.

Supplemental Information

Supplemental Information 1 DBP induced acute toxicity and oxidative damage in Daphnia magna.

Table 1 raw data is available in Sheet 1 (neonate acute) and Sheet 2 (adult acute). Figure 1 raw data is available in Sheet 3 (H2O2). Figure 2 raw data is available in Sheet 4 (MDA). Figure 3 raw data is available in Sheet 5 (T-AOC). Figure 4 raw data is available in Sheet 6 (SOD). Figure 5 raw data is available in Sheet 7 (CAT). Figure 6 raw data is available in Sheet 8 (GST).

Click here for additional data file.

Supplemental Information 2 Analytical determinations of the DBP using GC/MS.

Click here for additional data file.

Supplemental Information 3 statistical analysis of the data using SPSS.

Click here for additional data file.

Additional Information and Declarations

Competing Interests

Author Contributions

Data Availability

The authors declare that they have no competing interests

Chenchen Shen performed the experiments, analyzed the data, prepared figures and/or tables.

Jie Wei conceived and designed the experiments, contributed reagents/materials/analysis tools, approved the final draft.

Tianyi Wang performed the experiments.

Yuan Wang performed the experiments, analyzed the data, prepared figures and/or tables, authored or reviewed drafts of the paper, approved the final draft.

The following information was supplied regarding data availability:

The raw data is available as a Supplemental File.

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
