# Peer review of "Acute toxicity and responses of antioxidant systems to dibutyl phthalate in neonate and adult Daphnia magna"

_PeerJ, doi:10.7717/peerj.6584_

## Round 0.1 · original submission · Major Revisions

In your revisions, please address all reviewer comments. Please note that one of the reviewers included an annotated manuscript.

The following comments are particularly relevant.

1. How is it possible to get enough synchronized progeny from a single Daphnia in a single day?
2. Add some brief results on the stability of DBP in water.
3. Concern regarding the inclusion of H2O2 and T-AOC results in oxidative damage paragraph. Can ROS and non-enzymatic antioxidant measurements be considered as damage?
4. Why Excel for the statistical analyses?
5. Cite more of the previous work done with phthalates and Daphnia.
6. Please have your revised manuscript edited by a native English speaker.

Reviewer 1 ·

Basic reporting

No comment

Experimental design

Experiment is designed properly.

Validity of the findings

No comment

Additional comments

I think the manuscript is edited properly and it should be considered for publication. However, author should clarify one point regarding Daphnia culture.
It is mentioned that a single Daphnia was used to get progeny. How is it possible to get enough synchronized progeny from a single Daphnia in a single day?
It should be clarified how the experiment was setup.

Reviewer 2 ·

Basic reporting

The manuscript was written well, in clear English. Reference is updated and clearly reported the current background on teh investigated topic. Manuscript structure is ok, the data have been shared and are usable to repeat the analyses.

Experimental design

The research fits the aim of the Journal. The experiment is interesting, the tested molecule is of concern and limited information on its toxicity is curretly available. The experiment was performed well. As suggested by the reviewers of the first submission, the authors replicated the experiments to include a t=0 value that in the former manuscript missed. Adding the t=0 baseline data strenghten the results of the paper.

I suggest to add some brief results on the stability of DBP in water, although these data are reported in many previous papers.

I have a little concern about the paragraph oxidative stress status. Oxidative stress raises when oxidative status is unbalanced. So oxidative stress status in my opinion has no sense. I suggest to change the title of the paragraph in materials and metods section as: Oxidative stress assays (biomarkers). Moreover, in the next paragraph I suggest to include T-AOC because this assay measures the non-enzymatic antioxidant capacity. I have also a concern regarding the inclusion of H2O2 and T-AOC results in oxidative damage paragraph. The alteration on these parameters (i.e. the unbalance of the equilibrium between ROS and antioxidant in favor to the former) induces the oxidative stress, leading to oxidative damage. Thus, ROS and non-enzymatic antioxidant measurements cannot be considered as damage.

The applied statistical approach is correct (two-way ANOVA). However, I do not understand why you used Excel. What kind of analyses did you performe with Excel?

Validity of the findings

The results are interesting, data are robust (the number of experimental replicates is appropriate). Interpretation of results is correct and conclusions are generally based on results.

I have only one concern: both in the introduction and in discussion you stated that your results can help to find accurate biomarker to investigate DBP response in environment. In my opinion this is not possible because the suite of biomarker you applied is made by generalistic assays that respond to a plethora of contaminants. Thus, in environment, although you find an alteration in oxidative stress assays you cannot state that this effect was due to DBP rather than other pollutants. I should consider with care this issue within the manuscript.

Additional comments

Please check the common name of the species you referred within the text. Some common name is missing.

Please specify in the figure captions the meaning of capital (or lowercase) letters above the histograms.

Did you check (and report) for the time x treatment interaction effect?

Please include in the result section a Table reporting all the results of the statistical analysis, showing statistical indicator (F), degrees of freedom and P value. The P value is not enough. Alternatively, you can also report the information of statistics within the text (within brackets).

Reviewer 3 ·

Basic reporting

The article is clear and mostly unambiguous, however there are certain grammatical typos and language issues particularly in the results section (see specific comments below). Intro and background seem to be sufficient with the exception that the authors do not cite enough of the previous work done with phthalates and Daphnia. The novelty of this work is not the acute toxicity but is the linking of the acute toxicity data with the anti oxidant systems. The authors should stress this rather than the toxicity. The figures are relevant, good quality and sufficiently labelled and described however, some improvement is necessary for clarity.

Experimental design

The primary research is within the scope of the journal, the research question is well defined. The relevant and meaningfulness of this data increases our knowledge of effects phthalates, however the authors need to state a little more on how this will be used. The explanation of the research gap needs some additions. The research seems be done with reasonable technical and ethical standards.

The methods are mostly described in sufficient details and information to replicate, See specific comments below.

Validity of the findings

The findings appear meaningful, and the data appear to be robust statistically sound and robust.
The Conclusions are well stated, linked to original research question & limited to
supporting results. However the authors do not comment significantly on the Safe Concentration calculation in conclusion.

Additional comments

General comments
Introduction looks good.
Material and Methods:
The material and methods are good, but I would like a little more detail on each of the anti-oxidant methods as that is a very important part of this work.
Results:
Almost all first sentences of results paragraphs need to be rewritten or deleted. A few specific comments are below and indicated on PDF comments.
Line 255 – “To assed” should be corrected
Line 264 -265 – sentence sounds like part of the intro. Rather just report what was done.
Line 281 Line should be removed. (any sentences that just says “data was shown in figure ()” as its main info should be deleted.) See PDF

Line 392-395 See comment on PDF.

Table 1. Add column of how many trials and how many replicates per trial.

Figures: Please explain what comparisons are being made. It can be interpreted that the 24 h were compared with the 48 hours as stated now, but that does not look like it is the case.
Figure 1
A and B should have same y AXIS
Figure 2 – Explanation of the severe difference between adults and Neonates needs to be explained.
Figure 3
A and B should have same y AXIS
Figure 4
A and B should have same y AXIS
Figure 5
A and B should have same y AXIS
Figure 6
A and B should have same y AXIS

Annotated reviews are not available for download in order to protect the identity of reviewers who chose to remain anonymous.

---

## Round 0.2 · accepted · Accept

Thank you for your efforts in responding to reviewer comments and revising your manuscript. I appreciate also you taking the time to have the revised manuscript professionally edited.

#